# Zero-Shot Robustification of Zero-Shot Models With Auxiliary Foundation Models

## Abstract

Zero-shot inference is a powerful paradigm that enables the use of large pretrained models for downstream classification tasks without further training. However, these models are vulnerable to inherited biases that can impact their performance. The traditional solution is fine-tuning, but this undermines the key advantage of pretrained models, which is their ability to be used out-of-the-box. We propose ROBOSHOT, a method that improves the robustness of pretrained model embeddings in a fully zero-shot fashion. First, we use zero-shot language models (LMs) to obtain useful insights from task descriptions. These insights are embedded and used to remove harmful and boost useful components in embeddings—without any supervision. Theoretically, we provide a simple and tractable model for biases in zero-shot embeddings and give a result characterizing under what conditions our approach can boost performance. Empirically, we evaluate ROBOSHOT on nine image and NLP classification tasks and show an average improvement of 15.98% over several zero-shot baselines. Additionally, we demonstrate that ROBOSHOT is compatible with a variety of pretrained and language models.

## 1 Introduction

Zero-shot models are among the most exciting paradigms in machine learning. These models obviate the need for data collection and model training loops by simply asking the model for a prediction on any set of classes. Unfortunately, such models inherit biases or undesirable correlations from their large-scale training data [DLS+18, TE11]. In a now-canonical example [KSM+21], they often associate `waterbirds` with `water background`. This behavior leads to decreased performance, often exacerbated on rare data slices that break in-distribution correlations.

A growing body of literature [YNPM23, GKG+22, ZR22] seeks to improve robustness in zero-shot models. While promising, these works require labeled data to train or fine-tune models, and so **do not tackle the zero-shot setting.** A parallel line of research seeking to debias word embeddings [AZS+, BCZ+16, DP19, LGPV20] often sidesteps the need for labeled data. Unfortunately, these works often require domain expertise and painstaking manual specification in order to identify particular concepts that embeddings must be invariant to. As a result, out-of-the-box word embedding debiasing methods also cannot be applied to zero-shot robustification.

Can we robustify zero-shot models without (i) labeled data, (ii) training or fine-tuning, or (iii) manual identification? Surprisingly, despite this seemingly impoverished setting, it is often possible to do so. Our key observation is that zero-shot models **contain actionable insights** that can be exploited to improve themselves or other zero-shot models. These insights are noisy but cheaply available at scale—and can be easily translated into means of refinement for zero-shot representations. These refinements improve performance, particularly on underperforming slices—at nearly no cost.

Submitted to 37th Conference on Neural Information Processing Systems (NeurIPS 2023). Do not distribute.

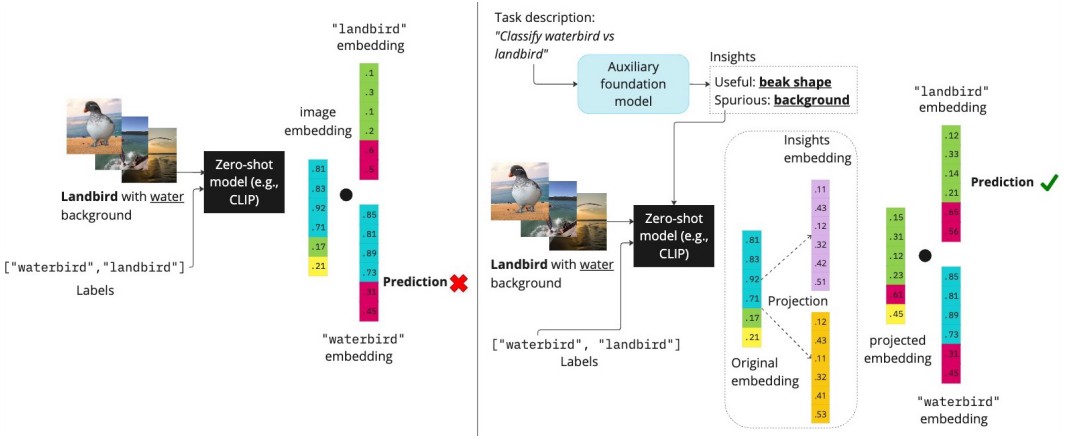

Figure 1: ROBOSHOT pipeline (right) vs. vanilla zero-shot classification (left).

We propose ROBOSHOT, a system that robustifies zero-shot models via auxiliary language models *without labels, training, or manual specification*. Using just the task description, ROBOSHOT obtains *positive and negative insights* from a language model (potentially the model to be robustified itself). It uses embeddings of these noisy insights to recover *harmful, beneficial*, and *benign* subspaces of zero-shot latent representation spaces. Representations are then modified to neutralize and emphasize their harmful and beneficial components, respectively.

Theoretically, we introduce a simple and tractable model to capture and quantify failures in zero-shot models. We provide a result that characterizes the *quantity and quality* of insights that must be obtained as a function of the severity of harmful correlations. Empirically, ROBOSHOT achieves 15.98% improvement across nine image and NLP datasets while offering sufficient versatility to apply to a diverse variety of base models. Most excitingly, in certain cases, it reaches comparable or greater improvements **even when compared to fine-tuned models** that rely on labeled data.

Our contributions include,

1. A simple theoretical model describing zero-shot model failures along with a theoretical analysis of our approach that characterizes the amount of information required for obtaining improvements as a function of the most harmful unwanted correlation,

2. ROBOSHOT, an algorithm that implements our core idea. It extracts insights from foundation models and uses them to improve zero-shot representations,

3. Extensive experimental evidence on zero-shot language and multimodal models, showing improved worst-group accuracy of 15.98% across nine image and NLP datasets.

## 2   Related Work

We describe related work in zero-shot model robustness, debiasing embeddings, guiding multi-modal models using language, and using LMs as prior information.

**Zero-Shot inference robustness.**   Improving model robustness to unwanted correlations is heavily studied [SKHL19, ABGLP19, KCJ+21, KIW22, LHC+21, LCT+22]. Some methods require training from scratch and are less practical when applied to large pretrained architectures. Existing approaches to improve robustness *post-pretraining* predominantly focus on fine-tuning. [YNPM23] detects spurious attribute descriptions and fine-tunes using these descriptions. Specialized contrastive loss is used to fine-tune a pretrained architecture in [GKG+22] and to train an adapter on the frozen embeddings in [ZR22]. While promising, fine-tuning recreates traditional machine learning pipelines (e.g., labeling, training, etc.), which contradicts the promise of zero-shot models. In contrast, our goal is to avoid any training and any use of labeled data.

**Debiasing embeddings.**   A parallel line of work seeks to de-bias text embeddings [AZS+] [BCZ+16] [DP19] [LGPV20] and multimodal embeddings [WZS22, BHB+22, WLW21] by re-

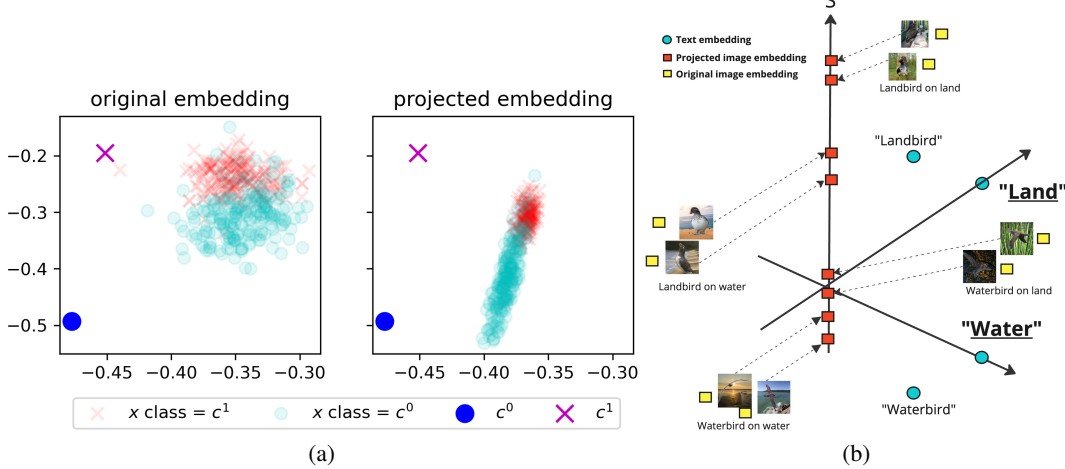

Figure 2: (a) ROBOSHOT debiases original input embedding (left). The projected embedding (right)'s variance in the unwanted direction is reduced, and in the relevant direction increases. (b) Embedding projection. We project embeddings to the space orthogonal to the embeddings of all unwanted insights (e.g., `water` and `land`)

moving subspaces that contain harmful or unwanted concepts. We use a similar procedure as a building block. However, these methods either target specific fixed concepts (such as gender) or rely on concept annotations, which limits their applicability across a wide range of tasks. In contrast, our method automates getting *both beneficial and unwanted concepts* solely from the task descriptions. An additional difference is that our goal is simply to add robustness at low or zero-cost; we not seek to produce fully-invariant representations as is often desired for word embeddings.

**Using language to improve visual tasks** A large body of work has shown the efficacy of using language to improve performance on vision tasks [RKH+21, FCS+13, LCLBC20]. Most relevant are those that focus on robustness, like [PDN+22], where attention maps using multimodal models (like CLIP) are used as extra supervision to train a downstream image classifier. [YNPM23] uses text descriptions of spurious attributes in a fine-tuning loss to improve robustness against spurious correlations. In contrast to these works, we focus on using textual concepts to improve zero-shot model robustness—without fine-tuning.

**Language model as prior** The basis of our work comes from the observation that language models contain information that can serve as a prior for other learning tasks. [KNST23] finds that LLMs can perform causal reasoning tasks, substantially outperforming existing methods. [CCSE22] explicitly prompts LLMs for task-specific priors, leading to substantial performance improvements in feature selection, reinforcement learning, and causal discovery. Our work shares the spirit of these approaches in using the insights embedded in language models to enhance zero-shot robustness.

## 3 RoboShot: Robustifying Zero-shot Models

We are ready to provide our setup and describe the algorithm.

### 3.1 Modeling and setup

Suppose that the zero-shot model's latent space contains an (unknown) *concept set*; similar notions have been studied frequently in the literature [DKA+]. For simplicity, we assume that this concept set is given by the orthonormal vectors $\{z_1, \dots, z_k\}$. The model's encoder produces, for a particular input a representation $x$ that is a mixture of concepts $\sum_i \gamma_i z_i$, where $\gamma_i \geq 0$ are weights.

We shall work with the following theoretical model for zero-shot classification. It closely resembles models like CLIP. For simplicity, we assume that there are two classes. It is straightforward to extend

**Algorithm 1:** ROBOSHOT

---

1: **Parameters:** Input embedding $x$, class embeddings $c^0, c^1$, harmful insight representations
   $v^1, \ldots, v^{|S|}$, helpful insight representations $u^1, \ldots, u^{|R|}$
2: **for** $j \in \{1, 2, \ldots, |S|\}$ **do**
3:     Reject harmful insight $v_j$: set $x \leftarrow x - \langle x, v^j \rangle / \langle v^j, v^j \rangle v^j$
4:     Renormalize $x = x / \|x\|$
5: **end for**
6: **for** $k \in \{1, 2, \ldots, |R|\}$ **do**
7:     Increase helpful insight $u_k$: set $x \leftarrow x + \langle x, u^k \rangle / \langle u^k, u^k \rangle u^k$
8: **end for**
9: $\hat{c} = \mathbb{1}\{x^T c^0 < x^T c^1\}$
10: **Returns:** Robustified zero-shot prediction $\hat{c}$

---

the analysis below to multiple classes. We take $\sum_i \alpha_i z_i$ to be the embedding of a datapoint, while $c^0 = \sum_i \beta_{i,0} z_i$ is the embedding of the first class and $c^1 = \sum_i \beta_{i,1} z_i$ is that of the second. Finally, we assume that we have access to $m$ answers $v^1, \ldots, v^m$ from the queries to the language model. These are given by $v^j = \sum_i \gamma_{i,j} z_i$ for $j \leq m$. We call these *insight representations*. Without our approach, the prediction is made by $\mathbb{1}\{(\sum_i \alpha_i z_i)^T (\sum_i \beta_{i,0} z_i) < (\sum_i \alpha_i z_i)^T (\sum_i \beta_{i,1} z_i)\}$, so that we predict whichever class has higher inner product with the datapoint's embedding.

Next, we assume that each input representation $x$ can be represented by partitioning the mixture components into three groups,

$$x = \sum_s^S \alpha_s^{\text{harmful}} z_s + \sum_r^R \alpha_r^{\text{helpful}} z_r + \sum_b^B \alpha_b^{\text{benign}} z_b.$$

The same holds for class and insight representations.

**Example**    We illustrate how harmful correlations produce errors on rare slices of data through a standard task setting, Waterbirds [KSM$^+$21]. In this dataset, the goal is to classify `landbirds` versus `waterbirds`, and the background (`land` or `water`) is spurious. Suppose that we have these terms relate to concepts such that $z_{\text{water}} = -z_{\text{land}}$ and $z_{\text{waterbird}} = -z_{\text{landbird}}$.

Consider a datapoint coming from a rare slice infrequently encountered in the training set. This might be an image of a landbird over water. Its embedding might be $x = 0.7 z_{\text{water}} + 0.3 z_{\text{landbird}}$. We may also have that

$$c_{\text{waterbird}} = 0.4 z_{\text{water}} + 0.6 z_{\text{waterbird}} \text{ and } c_{\text{landbird}} = 0.4 z_{\text{land}} + 0.6 z_{\text{landbird}}.$$

Then, $x^T c_{\text{waterbird}} = 0.1 > x^T c_{\text{landbird}} = -0.1$, so that the prediction is waterbird, and thus incorrect. This is caused by the presence of harmful components in *both* the class embedding (caused by seeing too many images with water described as waterbirds) and the datapoint embedding (where the water background appears). Thus our goal is to *remove* harmful components (the $z_s$'s) and *boost* helpful components (the $z_r$'s). We explain our approach towards doing so next.

### 3.2 ROBOSHOT: Zeroshot robustification with LLM

We describe ROBOSHOT in Algorithm 1. It uses representations of insights from language models to shape input and class embeddings to remove harmful components and boost helpful ones. Figure 2 is helpful in understanding the intuition behind these procedures. The left part (a) illustrates the effect of ROBOSHOT on a true dataset. Note how unhelpful directions are neutralized while others are boosted. The illustration on the right (b) shows this effect on the waterbirds running example.

**Obtaining insight representations from LMs**    The first question is how to obtain insight representations without training. To do so in a zero-shot way, we use *textual* descriptions of harmful and helpful concepts by querying language models using *only the task description*. For example, in the Waterbirds dataset, we use the prompt "What are the biased/spurious differences between waterbirds and landbirds?". We list the details of the prompts used in the Appendix. Let $s_1, s_2$ be the text insights obtained from the answer (e.g., {`water background`,' `land background`'}). We obtain a spurious insight representation by taking the difference of their embedding $v = \dfrac{g(s_1) - g(s_2)}{\|g(s_1) - g(s_2)\|}$,

where $g$ is the text encoder of our model.

In addition to attempting to discover harmful correlations, we seek to discover helpful components in order to boost their magnitudes past remaining harmful ones (or noise). The procedure is similar. We obtain insight representations using language models. For example, we ask "What are the true characteristics of waterbirds and landbirds?' and obtain e.g., {'short beak,' 'long beak'}. The remainder of the procedure is identical to the case of harmful components. Note that since we are seeking to boost (rather than remove) components, it is also possible to fix a multiplicative constant (to be treated as a hyperparameter) for the boosting procedure. That is, we could take $x \leftarrow x + \nu \times \langle x, u^k \rangle / \langle u^k, u^k \rangle u^k$ for some $\nu > 0$. While this is possible if we have access to a labeled set that we can tune $\nu$ over, we *intentionally avoid doing so to ensure our procedure is truly zero-shot*.

Prompting a language model is typically inexpensive, which will enable obtaining multiple insight vectors $\tilde{v}^1, \ldots, \tilde{v}^m$. From these, we obtain an orthogonal basis $v^1, \ldots, v^m$ separately for harmful and helpful components. Thus we have access to recovered subspaces spanned by such components.

**Removing and Boosting Components**   ROBOSHOT applies simple vector rejection to mitigate or remove harmful components, which is described in lines 2-5 of Algorithm 1. Similarly, it boosts helpful components as described in lines 6-9.

To see the impact of doing so, consider our earlier example. Suppose that $v^{\text{harmful}} = 0.9z_{\text{water}} + 0.1z_{\text{landbird}}$, and that this is our only harmful insight. Similarly, suppose that we obtain a single helpful insight given by $v^{\text{helpful}} = 0.1z_{\text{water}} + 0.9z_{\text{landbird}}$. Note that even these insights can be imperfect: they do not uniquely identify what are harmful or helpful concepts, as they have non-zero weights on other components.

We first obtain from removing the harmful component (ignoring normalization for ease of calculation) that

$$\hat{x} \leftarrow x - \frac{\langle x, v^{\text{harmful}} \rangle}{\langle v^{\text{harmful}}, v^{\text{harmful}} \rangle} v^{\text{harmful}} = -0.0244 z_{\text{water}} + 0.2195 z_{\text{landbird}}.$$

Then, we already we have that $x^T c_{\text{waterbird}} = -0.1415 < x^T c_{\text{landbird}} = 0.1415$, so that the correct class is obtained. In other words we have already, from having access to a single insight, neutralized a harmful correlation and corrected what had been an error. Adding in the helpful component further helps. We obtain

$$\hat{x} \leftarrow \hat{x} + \frac{\langle \hat{x}, v^{\text{helpful}} \rangle}{\langle v^{\text{helpful}}, v^{\text{helpful}} \rangle} v^{\text{helpful}} = -0.0006 z_{\text{water}} + 0.4337 z_{\text{landbird}}.$$

This further increases our margin. Note that it is not necessary to fully neutralize (i.e., to be fully invariant to) spurious or harmful components in our embeddings. The only goal is to ensure, as much as possible, that their magnitudes are reduced when compared to helpful components (and to benign components). In the following section, we provide a theoretical model for the magnitudes of such components and characterize the conditions under which it will be possible to correct zero-shot errors. We note that there is a variant of our approach that can also update class embeddings as well.

# 4   Analysis

Next, we provide an analysis that characterizes under what conditions ROBOSHOT is capable of correcting zero-shot errors. First, we consider the following error model on the weights of the various representations. For all benign representations, we assume that $\alpha_b, \beta_b, \gamma_b \sim \mathcal{N}(0, \sigma_{\text{benign}}^2)$. That is, the magnitudes of benign components are drawn from a Gaussian distribution. The value of $\sigma_{\text{benign}}$ is a function of the amount of data and the training procedure for the zero-shot model.

Next, we assume that the embedding insight $v_s = \sum_{i=1}^{k} \gamma_{i,s} z_i$ (where $1 \le s \le S$) satisfies the property that for $i \ne s$, $\gamma_{i,s} \sim \mathcal{N}(0, \sigma_{\text{insight}}^2)$, while $\gamma_{s,s}$ is a constant. In other words, the vectors $v_1, \ldots, v_S$ spanning the harmful component subspace are well-aligned with genuinely harmful concepts, but are also affected by noise. We seek to understand the interplay between this noise, benign noise, and the coefficients of the other vectors (i.e., helpful components). Let the result of rejecting embedding insights $v_1, \ldots, v_S$ be

$$\hat{x} = x - \sum_{s=1}^{S} \frac{x^T v_s}{||v_s||^2} v_s = \sum_i A_i z_i.$$

178 We provide a bound on $A_s$, the coefficient of a targeted harmful concept post-removal.

179 **Theorem 4.1.** *Under the noise model described above, the post-removal coefficient for harmful*
180 *concept $s$ satisfies*

$$|\mathbb{E}\left[A_s\right]| \leq \left|\frac{(k-1)\alpha_s\sigma_{insight}^2}{\gamma_{s,s}^2}\right| + \left|\sum_{t\neq s}^{S}\frac{\alpha_s\sigma_{insight}^2}{\gamma_{t,t}^2}\right|,$$

181 *where $k$ is the number of concepts.*

182 The theorem illustrates how and when the rejection component of ROBOSHOT works—it scales
183 down harmful coefficients at a rate inversely proportional to the harmful coefficients of the insight
184 embeddings. As we would hope, when insight embeddings have larger coefficients for harmful vectors
185 (i.e., are more precise in specifying terms that are not useful), ROBOSHOT yields better outcomes.
186 In addition, we observe that the harmful coefficients decrease when the insight embeddings have
187 less noise. In fact, we have that $\lim_{\sigma_{insight}\to 0} A_s = 0$ — the case of perfectly identifying harmful
188 concepts. In the Appendix, we present additional theoretical results for control of helpful coefficients
189 along with a combined result.

## 5 Experimental Results

191 This section evaluates the following claims about ROBOSHOT:

192 • **Improving multi-modal models (Section 5.1)**: ROBOSHOT improves zero-shot classification
193 robustness of various multi-modal models, even outperforming prompting techniques that include
194 spurious insight descriptions (which we do not have access to) in the label prompts.

195 • **Improving language models (Section 5.2)**: ROBOSHOT improves zero-shot robustness when
196 using language model embeddings for text zero-shot classification.

197 • **Extracting concepts from LM with varying capacities (Section 5.3)**: ROBOSHOT can extract
198 insights from language models with varying capacities. Improvements persist with weaker LMs.

199 • **Ablations (Section 5.4)** ROBOSHOT benefits from both removing harmful and boosting helpful
200 representations (line 3 and line 7 in ROBOSHOT Algorithm 1).

201 **Metrics and how to interpret the results.** We use three metrics: average accuracy % (AVG),
202 worst-group accuracy % (WG), and the gap between the two (Gap). While a model that relies on
203 harmful correlations may achieve high AVG when such correlations are present in the majority of the
204 test data, it may fail in settings where the correlation is absent. **A robust model should have high**
205 **AVG and WG, with a small gap between them**.

206 **Baselines** We compare against the following sets of baselines:

207 1. **Multimodal baselines**: We compare against: (i) vanilla zero-shot classification (**ZS**) and (ii)
208 zero-shot classification with group information (**Group Prompt ZS**). We do so across a variety of
209 models: CLIP (ViT-B-32 and ViT-L-14) [RKH⁺21], ALIGN [JYX⁺21], and AltCLIP [CLZ⁺22].
210 Group Prompt ZS assumes access to spurious or harmful insight annotations and includes them
211 in the label prompt. For instance, the label prompts for waterbirds dataset become [`waterbird`
212 `with water background, waterbird with land background, landbird with water`
213 `background, landbird with land background`]. We only report Group Prompt ZS results
214 on datasets where spurious insight annotations are available.

215 2. **Language model baselines**: We compare against zero-shot classification using multiple language
216 model embeddings, including BERT [RG19] and Ada [NXP⁺22] (**ZS**).

### 5.1 Improving multi-modal models

218

219 **Setup.** We experimented on five binary and multi-class datasets with spurious correlations and
220 distribution shifts, coming from a variety of domains: **Waterbirds** [SKHL19], **CelebA** [LLWT15],
221 **CXR14** [WPL⁺17], **PACS** [LYSH17], and **VLCS** [FXR13]. We use the default test splits of all
222 datasets. Dataset details are provided in the appendix. For CXR14, we use BiomedCLIP [ZXU⁺23],

Table 1: Main results. Best WG and Gap performance **bolded**, second best underlined.

| Dataset | Model | ZS | | | GroupPrompt ZS | | | **ROBOSHOT** | | |
|---|---|---|---|---|---|---|---|---|---|---|
| | | AVG | WG($\uparrow$) | Gap($\downarrow$) | AVG | WG($\uparrow$) | Gap($\downarrow$) | AVG | WG($\uparrow$) | Gap($\downarrow$) |
| Waterbirds | CLIP (ViT-B-32) | 80.7 | 27.9 | 52.8 | 81.6 | 43.5 | 38.1 | 82.0 | **54.4** | **28.6** |
| | CLIP (ViT-L-14) | 88.7 | 27.3 | 61.4 | 70.7 | 10.4 | 60.3 | 79.9 | **45.2** | **34.7** |
| | ALIGN | 72.0 | **50.3** | 21.7 | 72.5 | 5.8 | 66.7 | 50.9 | 41.0 | **9.9** |
| | AltCLIP | 90.1 | 35.8 | 54.3 | 82.4 | 29.4 | 53.0 | 78.5 | **54.8** | **23.7** |
| CelebA | CLIP (ViT-B-32) | 80.1 | 72.7 | 7.4 | 80.4 | 74.9 | 5.5 | 84.8 | **80.5** | **4.3** |
| | CLIP (ViT-L-14) | 80.6 | 74.3 | 6.3 | 77.9 | 68.9 | 9.0 | 85.5 | **82.6** | **2.9** |
| | ALIGN | 81.8 | 77.2 | 4.6 | 78.3 | 67.4 | 10.9 | 86.3 | **83.4** | **2.9** |
| | AltCLIP | 82.3 | **79.7** | **2.6** | 82.3 | 79.0 | 3.3 | 86.0 | 77.2 | 8.8 |
| PACS | CLIP (ViT-B-32) | 96.7 | 82.1 | 14.6 | 97.9 | 82.7 | 15.2 | 97.0 | **86.3** | **10.7** |
| | CLIP (ViT-L-14) | 98.1 | 79.8 | 18.3 | 98.2 | **86.6** | **11.6** | 98.1 | 83.9 | 14.2 |
| | ALIGN | 95.8 | **77.1** | **18.7** | 96.5 | 65.0 | 31.5 | 95.0 | 73.8 | 21.2 |
| | AltCLIP | 98.5 | 82.6 | 15.9 | 98.6 | 85.4 | 13.2 | 98.7 | **89.5** | **9.2** |
| VLCS | CLIP (ViT-B-32) | 75.6 | 20.5 | 55.1 | | - | | 76.5 | **33.0** | **43.5** |
| | CLIP (ViT-L-14) | 72.6 | 4.20 | 68.4 | | - | | 71.1 | **12.6** | **58.5** |
| | ALIGN | 78.8 | 33.0 | 45.8 | | - | | 77.6 | **39.8** | **37.8** |
| | AltCLIP | 78.3 | 24.7 | **53.6** | | - | | 78.9 | **25.0** | 53.9 |
| CXR14 | BiomedCLIP | 55.3 | 28.9 | 26.4 | | - | | 56.2 | **41.6** | **14.6** |

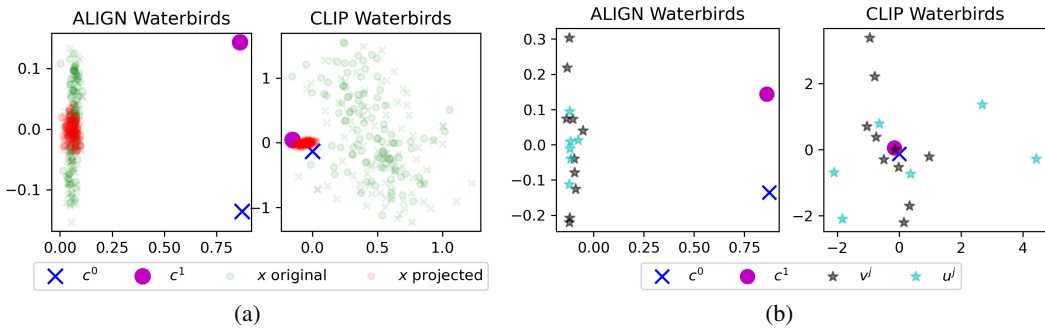

Figure 3: (a) Original (green) and projected (red) input embeddings $x$, and label embeddings $c^0$ and $c^1$. (b) label embeddings $c^0$ and $c^1$, harmful insight embeddings $v^k$ (black star) and helpful insight embeddings $u^j$ (blue star)

which is a variant of CLIP finetuned on biomedical images and articles. All experiments are conducted using frozen pretrained models.

**Results.** Table 1 shows that **ROBOSHOT significantly improves the worst group performance (WG)** and maintains (and sometimes also improves) the overall average (AVG) without any auxiliary information (in contrast to Group Prompt, which requires access to spurious insight annotation).

Improved robustness nearly across-the-board suggests that both the insights extracted from LMs and the representation modifications are useful. We also provide insights insights into the case where our method does not improve the baseline (ALIGN model on Waterbirds) in Fig. 3. In Fig. 3a, we visualize the original and projected input embeddings ($x$ in green and red points, respectively), and the label embeddings ($c^0$ and $c^1$). Fig. 3a (left) shows the embeddings from the ALIGN model. We observe that the projected embeddings (red) still lie within the original embedding space, even with reduced variance. In contrast, when examining the CLIP model embeddings (Figure 3a (right)), we observe that the projected embeddings are significantly distant from the original ones. Unsurprisingly, Figure 3b (left) reveals that $v^j$ and $u^k$ (harmful and helpful insight embeddings in black and blue stars, respectively) are not distinguishable in the text embedding space of ALIGN, collapsing the input embeddings after ROBOSHOT is applied.

Table 2: ROBOSHOT text zero-shot classification. Best WG in **bold**.

| Dataset | Model | ZS | | | ROBOSHOT | | |
|---|---|---|---|---|---|---|---|
| | | AVG | WG($\uparrow$) | Gap($\downarrow$) | AVG | WG($\uparrow$) | Gap($\downarrow$) |
| CivilComments | BERT | 48.1 | 33.3 | 14.8 | 49.7 | **42.3** | **7.4** |
| | Ada | 56.2 | 43.2 | 13.0 | 56.6 | **44.9** | **11.7** |
| HateXplain | BERT | 60.4 | 0.0 | 60.4 | 57.3 | **14.0** | **43.3** |
| | Ada | 62.8 | 14.3 | 48.5 | 63.6 | **21.1** | **42.5** |
| Amazon | BERT | 81.1 | 64.2 | 16.8 | 81.0 | **64.4** | **16.6** |
| | Ada | 81.2 | 63.4 | **17.8** | 82.9 | **63.8** | 19.1 |
| Gender Bias | BERT | 84.8 | 83.7 | 1.1 | 85.1 | **84.9** | **0.2** |
| | Ada | 77.9 | 60.0 | 17.9 | 78.0 | **60.1** | 17.9 |

Table 3: ROBOSHOT with LMs of varying capacity. Best WG **bolded**, second best underlined

| Dataset | ZS | | Ours (ChatGPT) | | Ours (Flan-T5) | | Ours (GPT2) | | Ours (LLaMA) | |
|---|---|---|---|---|---|---|---|---|---|---|
| | AVG | WG | AVG | WG | AVG | WG | AVG | WG | AVG | WG |
| Waterbirds | 80.7 | 27.9 | 82.0 | **54.4** | 72.1 | 32.4 | 88.0 | 39.9 | 84.8 | 36.5 |
| CelebA | 80.1 | 72.7 | 84.8 | 80.5 | 77.5 | 68.2 | 80.3 | 74.1 | 84.2 | **82.0** |
| PACS | 96.7 | 82.1 | 97.0 | **86.3** | 96.2 | 80.3 | 97.2 | 74.0 | 94.8 | 71.9 |
| VLCS | 75.6 | 20.5 | 76.5 | **33.0** | 69.6 | 20.5 | 75.5 | 26.1 | 72.0 | 18.2 |

## 5.2 Improving language models

**Setup.** We experimented on four text classification datasets: **CivilComments-WILDS** [BDS+19, KSM+21], **HateXplain** [MSY+21], **Amazon-WILDS** [NLM19, KSM+21] and **Gender Bias** classification dataset [DFW+20, MFB+17]. We use the default test splits of all datasets. In text experiments, the distinctions between harmful and helpful insights are less clear than for images. For this reason, we only use harmful vector rejection (line 3 in ROBOSHOT) in text experiments. CivilComments and HateXplain are toxic classification datasets with unwanted correlation between toxicity labels and mentions of demographics (e.g., male, female, mentions of religions). The datasets are annotated with demographic mentions of each text, and we directly use them to construct $v^j$. For Amazon and Gender Bias datasets, we query LMs with task descriptions. All experiments are conducted using frozen pretrained models.

**Results.** Table 2 shows that ROBOSHOT also improves zero-shot text classification in text datasets, as shown by our consistent boost over the baselines across all datasets.

## 5.3 Extracting concepts from LMs with varying capacities

**Setup.** We use LMs with different capacities: **ChatGPT** [OWJ+22], **Flan-T5** [CHL+22], **GPT2** [RWC+19], and **LLaMA** [TLI+23], to get harmful and helpful features insights ($v^j$ and $u^k$).

**Results.** Table 3 shows that ROBOSHOT can get insights on $v^j$ and $u^k$ from LMs of various capacities and improves zero-shot performance. Even though the the LM capacity correlates with the zero-shot performance, ROBOSHOT with weaker LMs still outperforms zero-shot (ZS) baseline.

## 5.4 Ablations

**Setup.** We run ROBOSHOT with only harmful component mitigation (reject $v^j$: ROBOSHOT line 3), only boosting helpful vectors (increase $u^k$: ROBOSHOT line 7), and both.

**Results.** The combination of both projections often achieves the best performance, as shown in Table 4. Figure 4 provides insights into the impact of each projection. Rejecting $v^j$ reduces variance in one direction, while increasing $u^k$ amplifies variance in the orthogonal direction. When both projections are applied, they create a balanced mixture. We note that when doing both projections does not

Table 4: Main results. Best WG and Gap performance **bolded**, second best underlined.

| Dataset | Model | ZS AVG | WG(↑) | Gap(↓) | Ours ($v^j$ only) AVG | WG(↑) | Gap(↓) | Ours ($u^k$ only) AVG | WG(↑) | Gap(↓) | Ours (both) AVG | WG(↑) | Gap(↓) |
|---|---|---|---|---|---|---|---|---|---|---|---|---|---|
| Waterbirds | CLIP (ViT-B-32) | 80.7 | 27.9 | 52.8 | 82.0 | 50.4 | 31.6 | 82.6 | 30.2 | 52.4 | 83.0 | **54.4** | **28.6** |
| | CLIP (ViT-L-14) | 88.7 | 27.3 | 61.4 | 82.7 | 35.8 | 46.9 | 88.3 | 29.8 | 58.5 | 79.9 | **45.2** | **34**.7 |
| | ALIGN | 72.0 | 50.3 | 21.7 | 56.4 | 41.6 | 14.8 | 62.8 | **56.4** | **6.4** | 50.9 | 41.0 | 9.9 |
| | AltCLIP | 90.1 | 35.8 | 54.3 | 81.4 | **59.0** | **22.4** | 89.1 | 35.2 | 53.9 | 78.5 | 54.8 | 23.7 |
| CelebA | CLIP (ViT-B-32) | 80.1 | 72.7 | 7.4 | 85.2 | **81.5** | **3.7** | 79.6 | 71.3 | 8.3 | 84.8 | 80.5 | 4.3 |
| | CLIP (ViT-L-14) | 80.6 | 74.3 | 6.3 | 85.9 | **82.8** | 3.1 | 80.0 | 73.1 | 6.9 | 85.5 | 82.6 | **2.9** |
| | ALIGN | 81.8 | 77.2 | 4.6 | 83.9 | 78.0 | 5.7 | 83.9 | 81.4 | **2.5** | 86.3 | **83.4** | 2.9 |
| | AltCLIP | 82.3 | **79.7** | **2.6** | 86.1 | 75.6 | 10.5 | 81.9 | 79.0 | 2.9 | 86.0 | 77.2 | 8.8 |
| PACS | CLIP (ViT-B-32) | 96.7 | 82.1 | 14.6 | 97.0 | 83.7 | 13.3 | 96.6 | 84.2 | 12.4 | 97.0 | **86.3** | **10.7** |
| | CLIP (ViT-L-14) | 98.1 | 79.8 | 18.3 | 98.0 | 79.8 | 18.2 | 98.1 | 83.8 | 14.3 | 98.1 | **83.9** | **14.2** |
| | ALIGN | 95.8 | 77.1 | 18.7 | 95.8 | **78.0** | **17.8** | 95.1 | 71.1 | 24.0 | 95.0 | 73.8 | 21.2 |
| | AltCLIP | 98.5 | 82.6 | 15.9 | 98.4 | 83.0 | 15.4 | 98.6 | 88.8 | 9.8 | 98.7 | **89.5** | **9.2** |
| VLCS | CLIP (ViT-B-32) | 75.6 | 20.5 | 55.1 | 75.6 | 22.7 | 52.9 | 76.4 | 29.5 | 46.9 | 76.5 | **33.0** | **43.5** |
| | CLIP (ViT-L-14) | 72.6 | 4.2 | 68.4 | 70.9 | 6.8 | 64.1 | 73.4 | 8.9 | 64.5 | 71.1 | **12.6** | **58.5** |
| | ALIGN | 78.8 | 33.0 | 45.8 | 78.2 | 30.7 | 47.5 | 78.0 | **43.2** | **34.8** | 77.6 | 39.8 | 37.8 |
| | AltCLIP | 78.3 | 24.7 | 53.6 | 77.5 | 24.4 | 53.1 | 79.0 | 20.5 | 58.5 | 78.9 | **25.0** | 53.9 |
| CXR14 | BiomedCLIP | 55.3 | 28.9 | 26.4 | 55.7 | **41.8** | **13.9** | 54.8 | 21.8 | 33.0 | 56.2 | 41.6 | 14.6 |

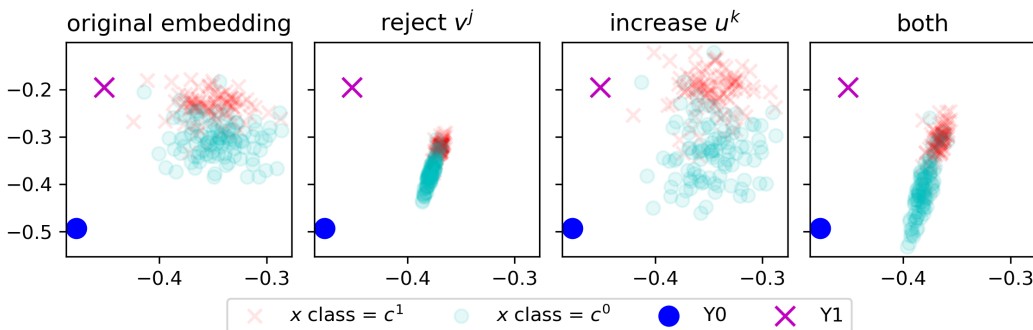

Figure 4: The effect of $v^j$ (reject), $u^j$ (increase), and both projections

improve the baseline, using only $u^k$ or $v^j$ still outperforms the baseline. For instance, the ALIGN model in the Waterbirds dataset achieves the best performance with only $u^k$ projection. This suggests that in certain cases, harmful and helpful concepts are intertwined in the embedding space, and using just one projection can be beneficial. We leave further investigation to future work.

## 6 Conclusion

We introduced ROBOSHOT, a fine-tuning-free system that robustifies zero-shot pretrained models in a truly zero-shot way. Theoretically, we characterized the quantities required to obtain improvements over vanilla zero-shot classification. Empirically, we found that ROBOSHOT improves both multi-modal and language model zero-shot performance, has sufficient versatility to apply to various base models, and can use insights from less powerful language models.

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
