# OpenReview forum: "Zero-Shot Robustification of Zero-Shot Models with Auxiliary Foundation Models"
_NeurIPS.cc/2023/Conference — Submitted to NeurIPS 2023_

### Official Review · Reviewer_Bg99 · 2023-06-11

**Soundness:** 3 good
**Presentation:** 3 good
**Contribution:** 3 good
**Rating:** 6
**Confidence:** 5

**Summary:**

This paper introduces RoboShot, a method that improves the robustness of pretrained model embeddings in a fully zero-shot manner. The key idea is to leverage insights obtained from language models based on task descriptions. These insights are used to modify the embeddings, removing harmful components and enhancing useful ones, without the need for supervision. Technically, the method ensures invariance to spurious features by projecting pretrained model embeddings onto the subspace orthogonal to the subspace spanned by spurious feature descriptions. Experiments demonstrate that RoboShot improves multi-modal and language model zero-shot performance.


**Strengths:**

1. **Novel and useful setting:** The setting of improving the robustness of pretrained model embeddings with task description is novel. RoboShot offers a unique approach that preserves the out-of-the-box usability of pretrained models, which is a key advantage.

2. **Extensive experiments and analyses:** The authors demonstrated the efficacy of the proposed method and setting with extensive experiments and analyses, in terms of both datasets and settings.

3. The paper is well-written.

**Weaknesses:**

1. **Limitation of method:** The robustification relies on the insights provided by language models. However, if the language model does not identify the potential failure cases of the model, the method cannot remedy it.

2. **Gender bias:** Some experiments are targeted at gender bias. It's better to discuss the scope of evaluation here, e.g., what genders are considered and what biases remain unresolved.



**Questions:**

1. What is the limitation of the proposed method, and when does it fail?

**Limitations:**

Limitation needs to be elaborated on.

---

> ### Author Rebuttal · Authors · 2023-08-09
>
>
> Thank you for noting the advantages of our paper:  **our novel setting and extensive experiments and analysis**! We also appreciate the helpful feedback.
>
>
> * **On gender bias experiments**. Thank you for pointing this out! We have added the details of genders considered in the dataset, as well as the remaining unresolved biases, to the updated Appendix of our updated draft. We have also added similar details for toxicity classification datasets.
>
>     We also briefly describe these additions: The gender bias dataset labels is not an exhaustive list of all genders. Only two genders are included in the dataset: male and female. Biases that might impact the gender demographics not included in the datasets labels remain unresolved.
>
>
> * **On limitations**. As mentioned in the common response, our theoretical framework provides the conditions to be effective. The bottom line---and key limitation---for our method to work is that the **insight noise should be sufficiently small compared to harmful/helpful coefficients**. To illustrate this point, we conducted a synthetic experiment varying the noise level in insight vectors. The following table shows the results. We see that up to 10~20% of noise level to signal (harmful, helpful coefficients = 1), our algorithm works well, recovering worst group accuracy and improving average group accuracy.
>
> |                                  | AVG            | WG             |
> | -------------------------------- | -------------- | -------------- |
> | Zero-shot                         | 74.90 $\pm$ 0.32  | 49.38 $\pm$ 0.83   |
> | RoboShot ($\sigma_{insight}=0.1$) | 96.80 $\pm$ 1.35  | 94.46 $\pm$ 0.29  |
> | RoboShot ($\sigma_{insight}=0.2$) | 96.38 $\pm$ 1.52    | 93.41 $\pm$ 3.36  |
> | RoboShot ($\sigma_{insight}=0.5$) | 89.50 $\pm$ 14.72  | 80.83 $\pm$ 21.45 |
> | RoboShot ($\sigma_{insight}=1$)   | 74.34 $\pm$ 24.51 | 55.83 $\pm$ 33.60 |
> | RoboShot ($\sigma_{insight}=5$)   | 49.56 $\pm$ 14.69 | 8.65 $\pm$ 12.61  |
>
> For completeness, we include the full set of experimental details below, and have added these to our draft.
> 1. Basis: $z_{core}=(1, 0, 0), z_{spurious}=(0,1,0), z_{benign}=(0, 0, 1)$
> 2. Class embeddings:
>     * $c_{1}=z_{core}+z_{spurious}+ z_{benign}$
>     * $c_{0}=-z_{core}-z_{spurious}+ z_{benign}$
> 3. Input distribution (here $s$ denotes spurious feature group):
>     * $x|y=1, s=0 \sim \mathcal{N}([w_{core}, w_{spurious}, w_{benign}], \sigma_{input}I), n=2500$
>     * $x|y=1, s=1 \sim \mathcal{N}([w_{core}, -w_{spurious}, w_{benign}], \sigma_{input}I), n=2500$
>     * $x|y=0, s=0 \sim \mathcal{N}([-w_{core}, -w_{spurious}, w_{benign}], \sigma_{input}I), n=2500$
>     * $x|y=0, s=1 \sim \mathcal{N}([-w_{core}, w_{spurious}, w_{benign}], \sigma_{input}I), n=2500$
> 4. Insight vectors:
>     * $v_{helpful} = \gamma_{helpful}z_{core} + \gamma_{s}z_{spurious} + \gamma_{b}z_{benign}$, where $\gamma_{s} \sim \mathcal{N}(0, \sigma_{inisght})$, $\gamma_{b} \sim \mathcal{N}(0, \sigma_{benign})$
>     * $v_{harmful} = \gamma_{c}z_{core} + \gamma_{harmful}z_{spurious} + \gamma_{b}z_{benign}$, where $\gamma_{c} \sim \mathcal{N}(0, \sigma_{inisght})$, $\gamma_{b} \sim \mathcal{N}(0, \sigma_{benign})$
> 5. For the experiment reported in Table, we used $w_{core}=1, w_{spurious}=1,  w_{benign}=0.5, \gamma_{helpful}=1, \gamma_{harmful}=1$, $\sigma_{input}=0.5, \sigma_{benign}=0.01$, and repetition=100

---

### Official Review · Reviewer_E4A7 · 2023-07-08

**Soundness:** 3 good
**Presentation:** 4 excellent
**Contribution:** 3 good
**Rating:** 6
**Confidence:** 4

**Summary:**

The paper presents an innovative approach to enhance zero-shot classification inference without the need for fine-tuning pre-trained models. The authors introduce a method that partitions input embeddings into three components: harmful, helpful, and benign. By leveraging task descriptions and querying language models with harmful and helpful prompts, they successfully extract harmful and helpful components. This leads to the removal of harmful components and a boost in the helpful ones, ultimately improving robustness in zero-shot classification. The proposed method demonstrates promise and contributes to the field of zero-shot classification without extensive model finetuning.

**Strengths:**

The paper is well-structured and easily understandable, with a strong and compelling motivation. As model sizes continue to increase, fine-tuning LLMs becomes increasingly expensive. This paper presents a compelling alternative by enhancing zero-shot performance while retaining the power of LLMs without the need for fine-tuning.  The innovative method of partitioning embeddings into three concepts and leveraging task descriptions and LLMs to strengthen or weaken them is intriguing and holds promise for embedding-based zero-shot text classification.

**Weaknesses:**

1. The proposed method is primarily applicable to embedding-based zero-shot classification approaches, while prompt-based methods like ChatGPT3.5/4 have gained popularity recently. Prompt-based methods allow humans to directly query language models based on downstream task knowledge, resulting in impressive performance. Although the proposed method is interesting for embedding-based zero-shot classification, its impact may be limited due to the current research trend.

2. Considering the above point, it would be beneficial if the authors compare their method to a more reasonable baseline, such as asking ChatGPT about predictions. This approach could be employed, for instance, by asking ChatGPT to identify if a given text contains gender bias. While ChatGPT's performance might not extend to image-based tasks, applying this baseline to text datasets would provide valuable insights. If ChatGPT achieves accurate predictions, it questions the necessity of an embedding-based zero-shot text classification method.

3. Although the method exhibits significant improvement in worst group performance (WG), it would strengthen the findings if the overall average performance also demonstrated improvement. It is worth noting that in some datasets, the average performance did not improve. This implies that the proposed method sacrifices performance for some classes to achieve better results in the WG. Ensuring a balance between overall average performance and WG improvement would bolster the method's effectiveness.

Overall, I am inclined to accept the paper if all weaknesses are properly addressed but I am also not strongly against rejecting the paper.

**Questions:**

1. The paper mentions that the average performance of some datasets is not improved. Could you please explain the possible reasons for this lack of improvement?

2. I'm curious about the performance of zero-shot text classification using ChatGPT. Can you provide any insights or comparative results on the performance achieved with ChatGPT for zero-shot text classification tasks?

3. In the paper, questions are posed to LLMs to extract helpful or harmful insights. Could you elaborate on how these questions are designed? Does it involve a significant amount of human decision-making and prior knowledge about the datasets?

**Limitations:**

Please refer to weaknesses.

---

> ### Author Rebuttal · Authors · 2023-08-09
>
> The reviewer notes that **our finetuning-free robustification method is innovative**. Thank you! We also appreciate the helpful input.
>
> * **On applicability to current research trends**. The **advantage of our approach is that it applies to and is capable of improving on prompt-based methods**! Indeed, just like prompting, it is fully zero-shot and does not require additional information. We need only access embeddings to turn a prompt-based method into one where our technique applies.
>
>     We validated this idea by following the reviewer's suggestion and conducting experiments by direct prompting to ChatGPT and BART-MNLI. The results of these experiments are in the common response. They show that **RoboShot improves direct prompting methods** on ChatGPT and BART, especially on the toxic classification tasks (25.7% in CivilComments and 9% in HateXplain).
>
> * **On average performance**. Thank you for the comment! Indeed, the fact that average performance is decreased when improving worst-group accuracy is a known phenomenon, which we discussed in the common response. We answer in more detail here. When removing harmful insights, RoboShot tries to remove spurious features which can be predictive for some groups (in-distribution data, e.g. water in waterbird images), but not across all groups (out-of-distribution data, e.g. water in landbird images). Thus, *accuracy in groups where spurious features were informative may drop slightly, while accuracy in groups where spurious features have adverse effects typically increases*. However, the tradeoff's appearance depends on the task, model, and embedding quality; **in many cases, average accuracy can substantially increase**. We note, as well, that this occurs only due to removal. When boosting helpful insights, our approach is beneficial to accuracy across all groups.
>
>     Thus, the case where average accuracy does not improve happens when 1) the removed harmful insights drop a group's accuracy in a way that outweighs the gains in other groups and 2) increasing helpful insights does not improve overall accuracy enough due to the embedding quality or weak helpful insights. Note that this tradeoff appears for the same reason even in fine-tuning based approaches, for example [1].
>
> * **On prompting LLMs to get textual insights**. The prompts are designed **solely based on the task descriptions** for each dataset. We described prompting approaches in the common response and refer to Tables 6 and 7 for the list of all prompts we use.
>
> [1] Zhang, Michael, and Christopher Ré. "Contrastive adapters for foundation model group robustness.", NeurIPS 2022.

---

> > ### Comment · Reviewer_E4A7 · 2023-08-17
> >
> > Thanks for the response and clarification. I keep my rating at 6.

---

### Official Review · Reviewer_DxCE · 2023-07-24

**Soundness:** 2 fair
**Presentation:** 2 fair
**Contribution:** 2 fair
**Rating:** 4
**Confidence:** 4

**Summary:**

This paper introduces a novel approach called ROBOSHOT that aims to enhance the robustness of pre-trained models for zero-shot classification tasks. The key idea is to leverage task-specific queries to prompt the large language model (LLM) to generate textual insights about the task, which can be classified as helpful or harmful concepts. These textual insights are then used to obtain "insight representations" based on the corresponding encoded embeddings, which are utilized to calibrate the vector of input representation to predict the class. The proposed method is evaluated through experiments on zero-shot image and text classification tasks. The paper also includes a theoretical analysis of the bound of the coefficient of targeted harmful concept post-removal.

**Strengths:**

1. The motivation of this paper is clear. The problem of "robustifying zero-shot models without labels, training, or manual specification" is challenging and interesting.
2. The paper proposes a simple method to calibrate the input embedding to make predictions.
3. The paper is generally easy to read and easy to follow. The settings of experiments in this paper are clear.
4. The theoretical proof of the coefficient bound of the targeted harmful concept is a plus.




**Weaknesses:**

1. The paper aims to robustify zero-shot models without labels, training, or manual specification. However, the proposed method still requires the use of manually-designed helpful/harmful queries (shown in Table 6&7) to query the LLM.
2. The paper did not explore the impact of using different prompts to query textual insights. Given that the experiments were conducted on small pre-trained models such as BERT, which are less robust than LLMs, it is unclear whether the resulting different text insights would significantly affect the model's performance.
3. The paper employs LLMs (e.g., chatgpt/LLAMA) to generate textual insights for calibrating the input embedding of a small pre-trained model for prediction. It raises the question of why not directly prompt the LLM for zero-shot text classification if it already possesses sufficient knowledge about each class. The paper would benefit from including such a baseline comparison.
4. The paper did not include and compare some relevant works, such as [1] [2], which calibrate the logits of predictions for zero-shot/few-shot text classification.
5. Table 1 indicates that the proposed method does not improve the ALIGN model on the Waterbirds dataset, but it does improve ALIGN's performance on CelebA, VLCS, and CXR14. The paper attributes this discrepancy to the harmful and helpful insight embeddings of Waterbirds being indistinguishable in the text embedding space of ALIGN. However, it is unclear how to determine when the ROBOSHOT method is applicable to different combinations of models and datasets. The paper could benefit from providing some metrics to predict the conditions under which ROBOSHOT is effective.

[1]Surface Form Competition: Why the Highest Probability Answer Isn’t Always Right

[2]Calibrate Before Use: Improving Few-Shot Performance of Language Models

**Questions:**

1. How can we determine when ROBOSHOT is suitable for different models and datasets?
2. Which dataset and model were used to generate the results presented in Figure 4, and what textual insights correspond to the "reject" and "increase" categories? Are there additional experiments or analyses that support the consistency of the proposed method's effectiveness with the analysis of the latent space?
3. Regarding lines 127-130, are the concise textual insights generated by the LLM manually determined by humans, or is there an automated process for deciding them? Given that the LLM is typically used to produce complete sentences, how is it adapted to generate these short and concise insights?

**Limitations:**

please refer to weakness point 5

---

> ### Author Rebuttal · Authors · 2023-08-09
>
> Thank you for the noting that our setting (model robustification without additional labels, training, and specification) is **challenging and interesting**! We also appreciate the helpful feedback.
>
> * **On the use of manually designed helpful/harmful queries to LLM**. Great question! The queries used to get the insights are solely based on the task description and **contain no manually tuned task-specific information**. We use prompts such as “list spurious/useful features to distinguish [classes]”. This is standard procedure to obtain answers from language models and akin to the practice of adding preceding text like “a photo of [label]” when prompting visual language models for zero-shot predictions.
>
>     To stay true to our zero-shot robustification settings, we **do not use any prompt tuning/extensive prompting engineering methods to obtain better performance** (doing so would likely boost the performance of our method even further). As seen in Tables 6 and 7, for stronger LLMs (ChatGPT, LLaMA) we simply ask them to *list* the spurious/useful features given a task. For GPT2 and FlanT-5 (no instruction tuning), we use several prompts (paraphrases of each other, mainly consisting of the class label string) to obtain a list of insights.
>
> * **On the impact of using different prompts to query textual insights**. We re-iterate our earlier point that we do not use any extensive prompt tuning, as the goal of our work  is to be truly zero-shot. This means we cannot tune prompts to obtain better performance. If we have access to a labeled dataset to be used for tuning, we could indeed obtain even better performance. However, the key insight in our work is that **substantial robustness improvements are possible without any** labeled data or additional tuning.
>
> * **On directly prompting LLM for classification**. This is also a great question! We address it in the common response. Briefly, RoboShot produces improvements in this comparison as well, by 25.7% in CivilComments and 9% in HateXplain. We have highlighted this result in the updated draft.
>
> * **On calibration**. We agree with the suggestion! Indeed, **RoboShot further benefits from the calibration methods** pointed out by the reviewer. In fact, this further highlights the versatility of Roboshot---we can combine it with such methods with no additional work. To showcase this, we show additional results below from (1) applying the calibration method alone, (2) our method, (3) the combination.
>
>     These new results show that **the best performing method across the board is either ours or the combination**. The underlying reason for this is that as the two methods are orthogonal, adding calibration can further improve the results.
> #### Model: BERT ####
> |Dataset|AVG|WG|
> |-|:-:|:-:|
> |CivilComments + Calibration|51.0|37.3|
> |CivilComments + **Ours**|49.7|**42.3**|
> |CivilComments + Combination|53.4|36.9|
> ||
> |HateXplain + Calibration|60.9|15.8|
> |HateXplain + **Ours**|57.3|14.0|
> |HateXplain + Combination|56.7|**22.8**|
> ||
> |Gender Bias + Calibration|85.4|83.2|
> |Gender Bias + **Ours**|85.1|**84.9**|
> |Gender Bias + Combination|85.7|82.5|
> ||
> |Amazon + Calibration|78.0|57.7|
> |Amazon + **Ours**|82.9|**63.8**|
> |Amazon + Combination|79.0|59.2|
>
> #### Model: Ada ####
> |Dataset|AVG|WG|
> |-|:-:|:-:|
> |CivilComments + Calibration|73.3|31.2|
> |CivilComments + **Ours**|56.6|**44.9**|
> |CivilComments + Combination|68.3|35.0|
> ||
> |HateXplain + Calibration|61.9|31.6|
> |HateXplain + **Ours**|63.6|21.2|
> |HateXplain + Combination|59.6|**33.3**|
> ||
> |Gender Bias + Calibration|84.2|77.8|
> |Gender Bias + **Ours**|78.0|60.1|
> |Gender Bias + Combination|84.2|**77.9**|
> ||
> |Amazon + Calibration|71.2|50.5|
> |Amazon + **Ours**|78.0|60.1|
> |Amazon + Combination|83.2|**63.9**|
>
>
> * **When is Roboshot applicable to different combinations of models and datasets.**
> This is an important question, and we have updated our draft with the answer. Briefly, there are two approaches to determining where RoboShot is suitable.
>     * _Intrinsic evaluation_: Our paper contains a theoretical framework characterizing our approach. We restate the results here: RoboShot scales down harmful coefficients in the sample embedding at an inversely proportional rate to the harmful coefficients in the insight embedding. This bound provides a key insight: that RoboShot works better when there is less noise in the insight embeddings. These quantities can then be directly measured on, e.g., synthetic data.
>     * _Extrinsic evaluation_: Existing pipelines built using zero-shot models are evaluated by backtesting or A/B testing. Since **our approach is a plug-in replacement for embeddings**, we can directly apply it to extrinsic evaluation techniques and compare.
>
>
> * **On Figure 4**. We use the CelebA dataset (sample 100 test points for each class), and use the real LLM insights. Some examples of the LLM (ChatGPT) outputs follow. Examples of spurious insights: [“deep complexion”, “ fair complexion”]. Examples of true insights: [“coarse hair texture”, “smooth hair texture”].
>
> * **On extracting concise textual insights from LLM output**. We fully automate the process of translating LLM outputs to concise textual insights. For ChatGPT, we instruct it to output in the format we can parse. For instance: “List the biased/spurious differences between waterbirds and landbirds. Give short keyword for each answer. Answer in the following format: <Difference>: <waterbird characteristic> ; <landbird characteristic>”. We can then directly parse ChatGPT’s output.  Other LLMs we use in the experiments are trained for the next word prediction task. So we take the outputs and slice the string by taking all characters after the prompt.

---

> ### Comment · Area_Chair_qmG7 · 2023-08-18
> **reminder**
>
> As the deadline is approaching, would you please check if the author reponse addresses your concerns? Thanks!

---

### Official Review · Reviewer_Kdvh · 2023-07-25

**Soundness:** 2 fair
**Presentation:** 3 good
**Contribution:** 2 fair
**Rating:** 4
**Confidence:** 3

**Summary:**

The main objective of this research is to enhance the robustness of image/text classification by taking into account the relationships between labels. The authors utilize Large Language Model (LLM) to incorporate prior knowledge about these labels. They also propose methods to amplify useful features and eliminate harmful features induced from these labels.

In contrast, prior work require manual identification of biased features for debiasing, whereas this work leverages LLM to automatically suggest such features, thereby reducing the need for manual effort.

**Strengths:**

- The paper is well written and can be easily understood.
- Using LLM to propose useful and spurious features is somewhat novel.

**Weaknesses:**

- This work aims to incorporate the knowledge of LLM into image classification. However, the method used, which is based on CLIP, does not seem to fully utilize the knowledge from LLM. The author's approach involves generating insight descriptions to establish a connection between LLM and CLIP's text encoder.  A more intuitive alternative would be to link CLIP's image encoder directly with LLM through an adapter, similar to the Flamingo (https://arxiv.org/abs/2204.14198) and LLaVA (https://arxiv.org/abs/2304.08485) methods. By doing so, the knowledge from LLM can be more comprehensively utilized, rather than solely the generated insight descriptions.
- The subspace-based debiasing methods have already been used under various contexts as also mentioned in the paper, which cannot be viewed as novel to some extent.

**Questions:**

- The multimodal baseline is good, how about a similar baseline that uses LLM to propose insight descriptions instead of using ground-truth? This gives better insights in comparing the benefit of injecting features in the image embedding side (e.g., by feature projection as introduced in this work) or the label side (e.g., by simply enriching the description).
- How about the proposed method benefits architectures like LLaVA and Flamingo?

**Limitations:**

The limitations part is missing in the paper.

---

> ### Author Rebuttal · Authors · 2023-08-09
>
> Thank you for noting the **novelty of our method**! We appreciate the valuable feedback.
> * **On other approaches, e.g., linking CLIP's encoder with LLMs with adapters**. This is a great idea! The challenge in doing so is that using adapters typically requires training on a dataset, while our setting **seeks to improve robustness without any further training, tuning, or extra labels**. Connecting a generative model to the vision encoder would need extra training and data (as in the suggested Flamingo and LLaVA combinations).
>
>     We note, however, that despite the fact that our setting is much more difficult, we can sometimes reach the same performance gains as scenarios where additional training data is possible. For example, the reviewer's suggestion of using adapters to tackle spurious features was pursued in [1]. While we face a more challenging setting, we _can achieve similar or superior performance gains_ in several cases.
>
> * **On novelty**. Indeed, as the reviewer notes, our novelty is not in the use of basic linear algebraic operations (the procedure that removes harmful subspaces that correspond to spurious correlations and strengthens the effect of helpful subspaces). Instead, the novelty of this work lies in:
>     * Our overall setting,
>     * Where and how we obtain the signal that permits these operations (querying language models),
>     * Our novel model that divides embeddings into harmful, helpful, and benign components, thus enabling us to seek to remove some and boost other components without any extra information or training.
>
>
> * **On using LLMs to propose insight descriptions**. We note that **we do not use any ground truth insights**. Using ground truth insights has been pursued in other work, e.g., [2]. Note that **our performance is superior to [2] on the common model used in both works (ViT-L/14), despite not accessing such ground truth insights!** Instead, we use insights obtained from LLMs to get textual descriptions of the spurious features for the task. We believe that this is exactly what is being suggested. At high level, our approach is to query LLMs via prompting, e.g., “what are spurious/useful features to distinguish [class X] and [class Y]?”. We then use embeddings of the answers to robustify base model embeddings.
>
>
>
> [1] Zhang, Michael, and Christopher Ré. "Contrastive adapters for foundation model group robustness.", NeurIPS 2022.
> [2] Chuang et al, "Debiasing Vision-Language Models via Biased Prompts", 2023.

---

> > ### Comment · Reviewer_Kdvh · 2023-08-19
> >
> > Thanks to authors for the response. I keep my rating.

---

> ### Comment · Area_Chair_qmG7 · 2023-08-18
> **reminder**
>
> As the deadline is approaching, would you please check if the author reponse addresses your concerns? Thanks!

---

### Author Rebuttal · Authors · 2023-08-09

$\newcommand{\ip}[2]{\left\langle#1, #2\right\rangle}$
We thank the reviewers for their kind comments and input. Before proceeding with in-depth responses, we highlight strengths of our work noted by reviewers.
* Our setting, **robustifying zero-shot models without extra labels and fine-tuning is novel and challenging** (DxCE, Bg99),
* We offer an **interesting alternative** as fine-tuning LLMs becomes increasingly costly. (E4A7),
* Our method is **simple** (DxCE) and **unique**. (Kdvh, E4A7, Bg99).

Given the novel nature of our work, the reviewers had several questions and suggestions they wished to see addressed before recommending acceptance. We appreciate these—and respond to all of them, often _producing further improved performance compared to the submitted version_**. We are confident that our work offers an exciting problem that invites further study—and a very strong baseline approach that has substantial practical value**.


We respond to three common questions.

* **On directly prompting LLMs for text classification** (Rev. DxCE and E4A7). As suggested, we conduct experiments on text datasets and compare them against the direct prompting of LLMs. In the following, we use ChatGPT and BART-MNLI. **RoboShot produces benefits here as well**, as large as a 25.7% improvement in CivilComments and 9% in HateXplain. Even in the Gender Bias experiments, RoboShot lifts weaker/older model performance to a level comparable to modern LLMs.

|Dataset+model|AVG|WG|
|-|:-:|:-:|
|CivilComments+ChatGPT|85.6|19.2|
|CivilComments+BART-MNLI|32.5|15.7|
|CivilComments+**Ours**(Ada)|56.6|**44.9**|
|CivilComments+**Ours**(BERT)|49.7|42.3|
||
|HateXplain+ChatGPT|55.4|12.2|
|HateXplain+BART-MNLI|61.2|5.3|
|HateXplain+**Ours**(Ada)|63.6|**21.2**|
|HateXplain+**Ours**(BERT)|57.3|14.0|
||
|Gender Bias+ChatGPT|90.1|**86.6**|
|Gender Bias+BART-MNLI|86.1|78.4|
|Gender Bias+**Ours**(Ada)|78.0|60.1|
|Gender Bias+**Ours**(BERT)|85.1|84.9|


* **On understanding improvements and limitations** (Rev. DxCE and Bg99). We agree! In fact, _capturing the conditions when improvements are or are not possible is the motivation for our theoretical analysis_. At high level, to be effective, we need that
    * Harmful insights should be aligned well with harmful concepts, i.e. should have large coefficients in our model in Sec. 3.1,
    * Helpful insights should be aligned well with helpful concepts and helpful concepts in class embeddings should have large coefficients,
    * Insight noise should be sufficiently small.

    In practice, these quantities are related to the quality of embeddings and the number and diversity of the language-model derived insights. To study the sensitivity to those factors, we varied embedding models and LLMs in our experiments. Our overall takeaway is that off-the-shelf modern pretrained models usually have enough of these properties to successfully robustify zero-shot prediction.

* **On average accuracy** (Rev. DxCE and Bg99). Improving worst-group accuracy leading to decreased average accuracy is a standard phenomenon, even when fine-tuning, e.g., see [1]. In that work, 2 out of 5 models have average accuracy drop when the worst group accuracy has been significantly improved.

    The underlying reason follows. Some inputs use non-causal (harmful) concepts as features, so that when this is blocked, average accuracy may decrease. Concretely, consider the distribution of the margin $M:\mathcal{X}\rightarrow \mathbb{R}$ given by  $M(x) := \ip{c^{+}}{x} - \ip{c^{-}}{x}$ where $c^+, c^-$ are the correct/incorrect class embeddings. Accuracy can be expressed as $\mathbb{E}[1[M(x)\geq 0]$. We observe this margin distribution in Figure A.1. (a) and Figure A.2. (a) in original image embeddings. Typically, inputs with spurious features ('waterbird'-'land') tend to be closer to the decision boundary ($M=0$). We expect that harmful insight removal procedure _increases_ the margin of $D_{sp}$, but _decreases_ the margin of inputs with non-spurious features $D_{nsp}$ (e.g. 'waterbird'-'water', which is shown in Figure A.1. (b) and Figure A.2. (b)). Here the potential **tradeoff between the accuracy of $D_{sp}$ and $D_{nsp}$ appears**. If the gain in $D_{sp}$ outweighs the loss in $D_{nsp}$, the average accuracy increases, as in Figure A.1.(b). If the gain in $D_{sp}$ is less than the loss in $D_{nsp}$, the average accuracy decreases. In either case, the model performance in $D_{sp}$ is improved by this procedure.

[1] Zhang, Michael, and Christopher Ré. "Contrastive adapters for foundation model group robustness.", NeurIPS 2022.

* **On designing LLM prompts to get textual insights** (Reviewers DxCE and E4A7). We **do not use any extensive prompt engineering or prompt tuning methods** in our paper, to stay true to our zero-shot setting. Concretely, for ChatGPT, we directly ask it to list the biased/spurious features and the true visual features for the task---regardless of the task. We also add instructions for the answer format, so we can automatically parse them. We use the same prompt used for LLaMA, without answer format instructions. We then parse the answer by slicing the string with indexes after the original prompt (so only take the answers to be extracted as insights). We use non-finetuned versions of GPT2 and Flan-T5, so we adapted the prompts for models trained on the next-word prediction task. We use several prompts that are paraphrases of each other, in order to get a list of insights from these models. The complete prompt list is in Tables 6 and 7.

---

> ### Author Response · Authors · 2023-08-11
>
> Dear Reviewers,
>
> We thank you again for your feedback, questions, and suggestions! We believe we have answered all of your questions in our responses and the updated draft. If you have additional questions, we would love to answer them!
>
> The Authors

---

### Decision · Program_Chairs · 2023-09-21

**Decision:**

Reject

**Comment:**

This paper presents RoboShot, a method for improving the robustness of pretrained model embeddings in zero-shot inference. However, reviewers have raised several concerns regarding the methodology, novelty, and experimental evaluation. The approach for incorporating knowledge from LLM into image classification seems suboptimal compared to recent methods like Flamingo and LLaVA. The debiasing techniques used are not viewed as novel, and the reliance on manually-designed queries undermines the claim of not requiring labels, training, or manual specification. Reviewers also question the impact of using different prompts and the absence of a baseline comparison with LLMs for zero-shot text classification. Reviewers also have conerns on that the paper does not provide clear guidance on when RoboShot is effective for different combinations of models and datasets. Due to these concerns, I recommend rejecting this paper in its current form.